# A Novel HIF Inhibitor Halofuginone Prevents Neurodegeneration in a Murine Model of Retinal Ischemia-Reperfusion

**DOI:** 10.3390/ijms20133171

**Published:** 2019-06-28

**Authors:** Hiromitsu Kunimi, Yukihiro Miwa, Hiroyoshi Inoue, Kazuo Tsubota, Toshihide Kurihara

**Affiliations:** 1Department of Ophthalmology, School of Medicine, Keio University, Shinjuku-ku, 160-8582 Tokyo, Japan; 2Laboratory of Photobiology, School of Medicine, Keio University, Shinjuku-ku, 160-8582 Tokyo, Japan; 3Department of Chemistry, School of Medicine, Keio University, Shinjuku-ku, 160-8582 Tokyo, Japan

**Keywords:** HIF-1α, halofuginone, ischemia-reperfusion, neuroprotection

## Abstract

Neurodegeneration caused with retinal ischemia or high intraocular pressure is irreversible in general. We have focused on the role of hypoxia-inducible factor (HIF) in retinal homeostasis and revealed that HIF inhibition may be effective against retinal neovascular and neurodegeneration. In this study, we performed in vitro screening of natural products and found halofuginone, which is a derivative of febrifugine extracted from hydrangea, as a novel HIF inhibitor. Administration of halofuginone showed a significant neuroprotective effect by inhibiting HIF-1α expression in a murine retinal ischemia-reperfusion model histologically and functionally. These results indicate that halofuginone can be a neuroprotective agent in ischemic retinal degenerative diseases.

## 1. Introduction

Among eye diseases, therapies for retinal neurodegeneration caused by acute ischemia or high intraocular pressure are limited. The neuronal damage in inner retina is irreversible in general—found in central retinal artery occlusion, ischemic optic neuropathy, or glaucoma. Therefore, development of an effective therapeutic approach is needed for such diseases. 

Hypoxia-inducible factor (HIF) plays an important role in cellular homeostasis especially under hypoxic condition induced by tissue ischemia. HIF is a transcription factor that regulates various gene expression to adapt to cellular hypoxia modulating neovascularization, glucose metabolism, proliferation, and apoptosis of the cells [1]. In case of the eye, ectopic expression and pathological contribution of HIF is found not only in acute ischemic [2] but also in chronically progressing neurodegenerative glaucomatous retinas [3]. In this study, we explore a novel HIF inhibitor to establish a therapeutic approach against inner retinal neurodegeneration while many of known HIF inhibitors such as topotecan and doxorubicin are concerned about systemic side effects [4,5]. 

In terms of safety, natural products may be ideal candidates to control cellular and tissue homeostasis. Recently, components obtained from foods and plants have been revealed to affect cellular system and gene expression [6,7]. To screen the biological functions of natural products, comprehensive analyses named nutrigenomics including PCR array, microarray, and luciferase assay may be performed [8]. Nutrigenomics are also leveraged in eye health care science [9]. Through such an approach, vitamin C and E is found to reduce the risk of age-related macular degeneration or cortical cataract as antioxidants since the oxidative stress has a pivotal role in retina and lens [10,11]. Recently, it has been reported that omega-3 fatty acid extracted from flaxseed oil suppressed diabetic retinopathy simulating G protein-coupled receptor 120 [12]. Recently, we screened 207 natural products utilizing a myopia suppressive gene *early growth response protein 1* (*EGR1*) as a biomarker and found out a carotenoid crocetin has a suppressive effect against progression of myopia in a murine model [13]. Taken together, the significance of natural compounds related to eye diseases has been widely recognized.

In the current study, we examined 238 natural products to discover HIF inhibitory agents. Through the screening analysis, halofuginone (7-bromo-6-chlorofebrifugine hydrochloride) was found to have an effect of inhibiting HIF activity in vitro and in vivo. It was reported that halofuginone, an analogue of febrifugine, had therapeutic effects in fibrosis and other pathological conditions [14]. However, effects of halofuginone in eyes have not been reported yet. In this study, we examined the neuroprotective effect of halofuginone in a murine model of retinal ischemia-reperfusion (I/R).

## 2. Results

### 2.1. Screening for Novel HIF Inhibitors

Our original library of natural compounds included 238 plants and food extracts. As the first screening, suppression of HIF activity was examined in the NIH/3T3 murine fibroblast cell line utilizing HIF luciferase dual assay. Cobalt chloride (CoCl_2_) was used to induce ectopic HIF activity and the suppressive effects of these products were evaluated (Table A1). Through the first screening, 78 candidates showed a stronger HIF suppressive effect compared to a positive control chemical topotecan. These compounds were further examined and statistically evaluated as the second screening (Table A2). 

### 2.2. HIF Inhibitory Effects of Halofuginone In Vitro

We found that hydrangea extracts had the strongest effect to inhibit HIF activation in the natural product library (Table A2). In this plant species, febrifugine is known as a component to have pharmacological activities while this alkaloid has some adverse reactions including nausea. Then, we examined the HIF inhibitory effect of halofuginone, which is a synthetic derivative of febrifugine with fewer side effects, in three different types of cell lines (NIH/3T3, 661W a murine cone photoreceptor cell line, and ARPE19 a human retinal pigment epithelium (RPE) cell line). Although it is ideal to use retinal ganglion cells (RGCs) for validating the effect of halofuginone in neurodegeneration of inner retina, currently there exists no available RGC cell line. Thus, we have used 661W and ARPE19 cell lines derived from the retina in this study. In all these cell lines, the increase of HIF activity induced by CoCl_2_ was significantly suppressed by administration of halofuginone (Figure 1). We further examined protein expression of HIF-1α induced by two different conditions of CoCl_2_ and actual hypoxia 1% oxygen. Western blotting showed that both CoCl_2_- and 1% hypoxia- induced HIF-1α expression in 661W and ARPE19 cells was suppressed by administration of halofuginone in a dose-dependent manner (Figure 2). These results indicated that halofuginone inhibited ectopic protein expression of HIF-1α in pseudo and real hypoxic conditions.

### 2.3. Halofuginone Inhibits Retinal HIF-1α and Target Genes In Vivo

Next, we examined the HIF inhibitory effect of halofuginone in vivo by intraperitoneal administration in the murine retinal I/R model. The increased HIF-1α protein expression in post-I/R retinas was significantly (*p* = 0.02) suppressed in halofuginone-treated mice compared to controls (Figure 3A,B). The upregulated retinal hif-1α and the target genes except for vegfa-a in post-I/R retinas were significantly suppressed in treated mice compared to controls (hif-1α: *p* = 0.028, vegf-a: *p* = 0.084, glut1: *p* = 0.019, pdk1: *p* = 0.026, respectively) (Figure 3C). These results suggested that administration of halofuginone inhibited increased HIF-1α and upregulated target gene expression in post I/R retinas.

### 2.4. Improvement of RGC Survival with Halofuginone Administration in Post-I/R Retinas

To assess the histological change of retinas after I/R injury, total retinal thickness was evaluated by optical coherence tomography (OCT). Total retinal thickness got thinner after I/R while those changes were prevented in halofuginone-treated mouse retinas (Figure 4A,B). We also evaluated the thickness with hematoxylin and eosin (H&E) stained retinal sections (Figure 4C). As consistent with the result in OCT, changes of the total retinal thickness were prevented by halofuginone administration (Figure 4D). The thickness of inner retinal layers between the inner retinal surface and the inner plexiform layer was significantly decreased with I/R damage. In contrast, the changes were suppressed in halofuginone-treated mouse retinas (Figure 4E). Transient high intraocular pressure (IOP) and ischemia in inner retina induced retinal ganglion cell (RGC) degeneration. To assess the protective effect of halofuginone for RGCs, we used fluorogold retrograde labeling. The decreased number of stained RGC nuclei observed in post-I/R retinas were significantly (*p* = 0.03) prevented in halofuginone-treated mouse retinas (Figure 5). These data indicated that halofuginone prevented RGC degeneration in the I/R model. 

### 2.5. Halofuginone Administration Protects I/R-Damaged Retina Electrophysiologically

We used electroretinography (ERG) to assess the neuroprotective effect of halofuginone against retinal I/R damage. The amplitudes of a- and b-wave in three conditions of ERG (rod, mix, and cone) were evaluated (Figure 6A,B). In rod a-wave (Figure 6C) and mix a- (Figure 6D) and b-wave (Figure 6E), decreased amplitudes were significantly (*p* < 0.05) prevented by halofuginone administration although the amplitude in cone b-wave was not changed by I/R (Figure 6F). Furthermore, we examined visual evoked potential (VEP) recordings (Figure 7A,B). The decreased amplitudes (Figure 7C) and extended latencies (Figure 7D) after I/R injury were also significantly (*p* = 0.047, 0.006) improved in the halofuginone-treated group. These results suggested that halofuginone had a neuroprotective effect functionally against I/R damage. 

## 3. Discussion

In this study, we found that the extracts from hydrangea had a HIF inhibitory effect among over 200 natural products (Table A2). In the hydrangea extracts, febrifugine was the possible substance to suppress ectopic HIF activities. Since febrifugine was known to have systemic side effects including nausea and vomiting, we investigated the HIF inhibitory effect of its synthetic derivative halofuginone. Halofuginone suppressed HIF-1α expression in two different conditions in a dose-dependent manner (Figure 2). In the murine retinal I/R model, increased HIF-1α expression and the upregulated target genes in the retina were prevented by systemic halofuginone administration (Figure 3). The histological examinations revealed that halofuginone treatment prevented the decrease of total retinal thickness and RGC number ( Figure 4; Figure 5). Electrophysiologically, the decrease of amplitudes in ERG and the prolonged latencies in VEP after the I/R injury were prevented in harofuginone-treated mice (Figure 6 and Figure 7). Taken together, the current data indicated that halofuginone is a novel HIF inhibitor showing a neuroprotective effect against RGC degeneration.

Halofuginone is one of the analogues of febrifugine isolated from Dichroa febrifuga [14]. Febrifugine was used to treat malaria fever in China over hundreds of years. Although febrifugine may have some side effects as oral treatment, halofuginone is more reasonable as a therapeutic agent. It has been reported that halofuginone suppressed skin fibrosis in vitro and in vivo [15,16]. This repressive response for collagen production may be induced by the suppression of TGF-β-dependent Smad3 phosphorylation [15]. Halofuginone administration also controls Th17 cell differentiation suppressing the progression of fibrosis [17]. In some animal models of muscular dystrophies, halofuginone treatments have shown a structural change of muscular tissue and a functional improvement [18]. Moreover, in the field of tumor therapy, halofuginone can reduce hepatoma cell tumor growth in a murine model [19,20] as well as other studies reporting suppressive effects of tumor such as prostate tumor, von Hippel-Lindau pheochromocytoma, and Wilm’s tumor [21,22,23]. For the use in clinical therapy, the safety of halofuginone was confirmed in a Phase I and II trials [24], thus halofuginone may be a strong candidate as an HIF inhibitor to clinical application.

In this study, we screened the candidate from natural products such as food and plant extracts. As well, many studies have exhibited the efficacy of foods for neuroprotection. For instance, the extracts from green tea, a Japanese traditional drink, improved brain dysfunction induced by high fat diet in senescence-accelerated mouse prone-8 (SAMP8) mice. Brain histological change, such as amyloid β accumulation or the reduction of brain-derived neurotrophic factor (BDNF), was suppressed with the diet including this extract [25]. In a primary culture of murine cortical neurons, administration of polyphenol-enriched extracts induced epigenetic change resulting in the cellular protection [26]. For Alzheimer’s disease (AD) and Parkinson’s disease (PD), three nuts (almond, hazelnut, and walnut) may have an effect to improve the cognitive function via suppressing the brain oxidative stress and amyloidogenesis [27]. It has been reported that macuna seed extracts, natural sources of Levodopa, decrease neurotoxin inducing neuroprotection [28]. Ellagitannins extracts from pomegranate flesh change the activity of protein disulfide-isomerase A3 (PDIA3) protecting the brain tissue from neurodegeneration involved in AD or PD [29]. Additionally, fatty acids in foods, especially omega-3 and -6, prevented RGC death caused by optic neuritis in a murine experimental autoimmune encephalomyelitis model [30]. A well-known soy component, isoflavones, reduce the oxidative stress in hippocampus in scopolamine-induced memory impairment mouse model [31]. These natural products and foods are possible candidates for neurodegenerative diseases.

Hence, it has been revealed that substances from foods or extracts of plants have neuroprotective effects, while the acting points have been unveiled. With nutrigenomics, safe products for oral intake may be possible drugs for neurodegenerative diseases. In the current study, we found that the extracts from hydrangea had a HIF inhibitory effect, then we showed that halofuginone is effective against RGC degeneration. The concept of nutrigenomics may have a more important role in the development of neuroprotective therapy.

## 4. Materials and Methods

### 4.1. Luciferase Assay for Drug Screening

The luciferase assay was performed as we previously described [32]. Murine fibroblast cell line NIH/3T3, murine cone photoreceptor cell line 661W, and human RPE cell line ARPE19 were transfected HIF-luciferase reporter gene construct (Cignal Lenti HIF Reporter, Qiagen, Venlo, Netherlands) to monitor HIF transcriptional activity. The HIF-luciferase construct encodes firefly luciferase gene under the control of hypoxia response element which binds HIFs. These cells were also co-transfected with CMV-renilla luciferase construct as an internal control. CoCl_2_ (200 μM, cobalt (II) chloride hexahydrate, Wako, Japan) was administrated to the cells in order to induce normoxic HIF activation 24 h before measuring the luminescence. To evaluate the suppressive effect of samples against HIF activation, 238 samples from the library of natural compounds were administrated at the same time as CoCl_2_ was added. The luminescence was measured with Dual-Luciferase^®^ Reporter Assay System (Promega, Fitchburg, WI, USA).

### 4.2. Western Blotting

Samples were homogenized in a lysis buffer (RIPA buffer (Thermo Fisher Scientific, Waltham MA, USA) containing protease inhibitor cocktail (Roche Switzerland). Protein lysates were boiled in Sodium dodecyl sulfate (SDS) loading buffer. The protein concentration was determined by using a bicinchoninic acid assay kit (Pierce, Rockford, IL, USA). Samples were fractionated on 10% SDS-polyacrylamide gels, transferred to nitrocellulose membranes, blocked with 5% nonfat dry milk, and incubated with primary antibodies mouse anti-HIF-1α (1:1500; #36169, CST, Danvers, MA, USA) or mouse anti-β-actin (1:4000; #A5316, Sigma-Aldrich, St Louis, MO, USA) at 4 °C overnight. Membranes were washed in tris buffered saline with Tween 20 (TBS-T) three times and incubated with horseradish peroxidase-conjugated secondary antibody goat anti-rabbit IgG (1:4000; #NA934, GE Healthcare, USA) or sheep anti-mouse IgG (1:4000; #NA931, GE Healthcare, Princeton, NJ, USA) in 5% nonfat dry milk for 1 h at room temperature and visualized using an ECL kit (Ez WestLumi plus, ATTO, Tokyo, Japan). Blotting was quantified using NIH ImageJ software.

### 4.3. RNA Extraction and qPCR

Total RNA was isolated from samples using TRI reagent (#TR118, MRC Global, Charleston, WV, USA) and Econospin column for RNA (#EP-21201, GeneDesign, Osaka, Japan). Columns were washed with Buffer RPE and RWT (#1018013, #1067933, Qiagen, Netherlands). After the extraction of RNA, RNA was reverse-transcribed into cDNA using ReverTra Ace qPCR RT Master Mix and qDNA remover (#FSO-301, TOYOBO, Osaka, Japan). RT-PCR was performed using THUNDERBIRD SYBR qPCR Mix (#QPS-201, TOYOBO, Japan) with StepOnePlus Real-Time PCR system (Applied Biosystems, Waltham, MS, USA). The 2^∆∆Ct method was used to calculate the relative amplification of cDNA fragments. Sequences of the primers were as follows: *hif-1α* forward; 5′-CCTGCACTGAATCAAGAGGTTGC-3′, *hif-1α* reverse; 5′-CCATCAGAAGGACTTGCTGGCT-3′, *vegf-a* forward; 5′-CTGCTGTAACGATGAAGCCCTG-3′, *vegf-a* reverse; 5′-GCTGTAGGAAGCTCATCTCTCC-3′, *pdk1* forward; 5′-GGCGGCTTTGTGATTTGTAT-3′, *pdk1* reverse; 5′-ACCTGAATCGGGGGATAAAC-3′, *glut1* forward; 5′-CAGTTCGGCTATAACACTGGTG-3′, glut1 reverse; 5′-GCCCCCGACAGAGAAGATG-3′, *gapdh* forward; 5′-AGGAGCGAGACCCCACTAAC-3′, *gapdh* reverse; 5′-GATGACCCTTTTGGCTCCAC-3′.

### 4.4. Animals

All procedures related to animal experiments were approved by the Institutional Animal Care and Use Committee of Keio University, and were in accordance with the National Institutes of Health (NIH) guidelines for work with laboratory animals, the Association for Research in Vision and Ophthalmology (ARVO) statement for the Use of Animals in Ophthalmic and Vision Research, and Animal Research: Reporting of in Vivo Experiments (ARRIVE) guidelines. All experiments were performed with 8-weeks-old male C57/BL6J mice (CLEA Japan, Yokohama, Japan). Animals were divided into two groups randomly in the completely blind manner and intraperitoneally injected with phosphate buffered saline (PBS) or halofuginone dissolved in PBS (0.2 mg/kg, Sigma-Aldrich, USA) once per day for 7 days. All mice were bred with a standard rodent diet (MF, Oriental Yeast Co., Ltd., Japan) and given free access to water. All cages were maintained under controlled lighting conditions (12 h light/12 h dark).

### 4.5. Retinal Ischemia/Reperfusion (I/R) Injury

Various murine retinal I/R techniques with different IOP and time have been addressed [33,34], while in the current study, the method with IOP of 90–99 mmHg in 30 min was adopted. Mice were anesthetized using intraperitoneal injection of medetomidine (0.75 mg/kg, Sandoz K.K., Tokyo, Japan), midazolam (4 mg/kg, Domitor^®^, Orion Corporation, Espoo, Finland) and butorphanol tartrate (5 mg/kg, Meiji Seika Pharma Co., Ltd., Japan) dissolved in normal saline (MMB). The mydriatic agent (Tropicamide, Phenylephrine Hydrochloride, Santen Pharmaceutical Co., Ltd., Osaka, Japan) eye drop was added to the eye. Cannulation of the anterior chamber was performed to increase IOP to 90–99 mmHg by infusion of normal saline into the eye with 35-gauge stainless needle. IOP was checked with a tonometer (TonoLab, Icare Finland Oy, Vantaa, Finland) every 5 min. The High IOP condition was maintained for 30 min, then the needle was pulled out. 

### 4.6. Evaluation of Retinal Thickness

Mouse total retinal thickness was evaluated using an OCT system (Envisu R4310, Leica, Wetzlar, Germany). Mice were anesthetized with MMB and fixed in the system. Ten points of whole retinal thickness (250 μm radius and every 36° centered upon the optic disc) were measured. The average number of them was evaluated as the value of the eye.

For H&E staining, eyes were enucleated from anesthetized mice, and frozen in the O.C.T. compound (Sakura Finetek Japan, Tokyo, Japan). Samples were sectioned by a cryostat (CryoStar, Thermo Fisher Scientific, USA). Slides were fixed in methanol for 5 min and dipped in hematoxylin solution for 90 s followed by a wash in water for 15 min. Then, slides were dipped in eosin solution for 30 s, and dipped 5 times in 100% ethanol and 3 times in Clear Plus (Falma, Tokyo, Japan) subsequently. Once slides were dried, mounting medium was applied and the slides were cover-slipped. Two points of retinal thickness (1 mm from the optic disc) were measured in a section, and the averages of them were evaluated as the value of an eye.

### 4.7. Retrograde Labeling of Retinal Ganglion Cells (RGCs)

Mice were anesthetized using MMB and fixed on the stereotaxic apparatus. The hair was shaved, and the scalp was cut along the parietal midline. The hole for injection was made 4.5 mm behind the bregma along the anteroposterior axis and 0.5 mm lateral to the midline using a φ0.5 mm stainless drill. 2 μL of 4% fluorogold (#SC358883, Santa Cruz, CA, USA) was injected through the hole using a Hamilton syringe. The injection was performed at a depth of 2 mm from brain surface and completed over 3 min. Seven days after the injection, mice were anesthetized, then each eye was enucleated. Eyes were fixed in 4% paraformaldehyde (PFA) at 4 °C for 15 min. The retina was dissected, quartered, and flat-mounted on a glass slide. Labeling RGCs were observed using BZ-9000 (KEYENCE Co, Osaka, Japan). RGCs were counted in 200 μm square at 1 mm distance from the optic nerve head. Four different squares in each quarter were counted, and the average of them was evaluated.

### 4.8. Electrophysiological Examination

In order to perform VEP recordings, two stainless steel pan-head screws (M1.0 × 6.0 mm) used for electrodes were inserted 1.0 mm in depth into the skull with hair shaved on the scalp around under MMB anesthesia before the I/R injury. These screws were placed over the bilateral primary visual cortex (1.5 mm anterior and 1.5 mm lateral to lambda). Mice were dark-adapted overnight before the electrophysiological examination. Full field ERG and VEP were recorded at 7 days after I/R using an acquisition system (PuREC, Mayo, Japan) with an LED stimulator in a Ganzfeld dome. Mice were anesthetized and mydriatic agent was dropped to both eyes. For ERG recordings, custom active electrodes were placed with contact lens (Mayo, Japan) and a reference electrode was placed in mouth. Three ERG stimulation intensities were used throughout recordings; 0.02 cds/m² 4 times for rod, 50 cds/m² one time for mix, and 20 cds/m^²^ 32 times for cone conditions. For VEP recordings, the active electrodes were inserted into the screws of skull and the reference electrode was placed in the mouth. A clipping electrode to the tail served as a ground. All recordings were measured in photopic adaptation status with 1 Hz white flashlights (3 cds/m²). Responses were arithmetically averaged with up to 64 records. The signals were 25 Hz low pass filtered. Each animal was recorded twice for both eyes. The amplitudes were taken with the first negative peak (P1-N1).

### 4.9. Statistical Analysis

The data were presented as the mean ± the standard deviation. Comparison of two experimental conditions was evaluated using unpaired Student’s *t*-test. A *p* < 0.05 was considered statistically significant.

## 5. Conclusions

We found a novel HIF inhibitor halofuginone screened from 238 natural products in our original library. Halofuginone has a neuroprotective effect against retinal I/R damage. The current study indicates that halofuginone is a possible drug for RGC degeneration induced by ischemia or high IOP inhibiting HIF-1α in the retina.

## 6. Patents

The current data includes patents applied by Keio University for a therapeutic or prophylactic agent for ischemic disease, glaucoma, optic nerve disease, retinal degenerative disease, angiogenic retinal disease, cancer, neurodegenerative or autoimmune disease, and a hypoxia inducing factor inhibitor (application no. PCT/JP2017/040884).

## Figures and Tables

**Figure 1 ijms-20-03171-f001:**
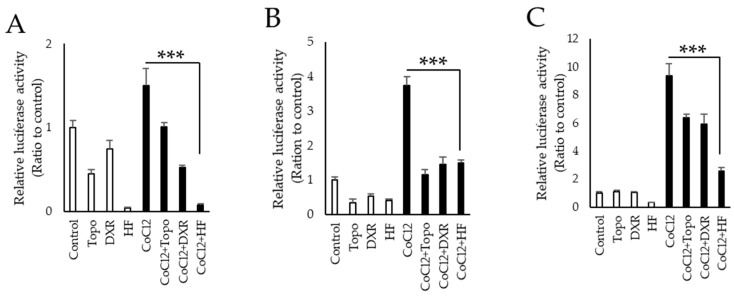
Hypoxia-inducible factor (HIF) inhibitory effects of halofuginone in vitro. HIF-reporter luciferase assay was performed in NIH/3T3 (**A**), 661W (**B**), and ARPE19 (**C**) cell lines (*n* = 3). Topotecan, doxorubicin, and halofuginone (100 μM) were administrated in normal or CoCl_2_-induced cultured condition. Note that halofuginone inhibited CoCl_2_-induced HIF activities stronger than two known HIF inhibitors. Error bars indicate the standard deviation. HF; halofuginone, Topo; topotecan, DXR; doxorubicin, CoCl_2_; cobalt chloride. *** *p* < 0.001, Student’s *t*-test.

**Figure 2 ijms-20-03171-f002:**
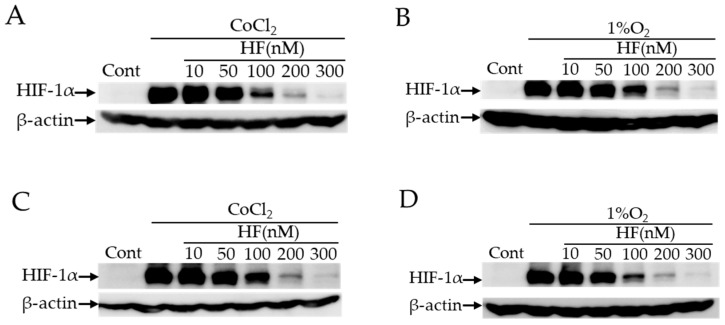
Inhibition of HIF-1α protein expression by halofuginone administration in 661W and ARPE19 cells. Western blotting for HIF-1α under CoCl_2_ (**A**) or 1% oxygen conditions (**B**) in 661W cells, and under CoCl_2_ (**C**) or 1% oxygen conditions (**D**) in ARPE19 cells. CoCl_2_ was administrated at the concentration of 200 μM, halofuginone was added at 10, 50, 100, 200, or 300 nM simultaneously, and cells were incubated for 6 h. The hypoxic condition was maintained in 1% O_2_ for 6 h. Halofuginone inhibited HIF-1α protein expression in a dose-dependent manner in both conditions. HF; halofuginone.

**Figure 3 ijms-20-03171-f003:**
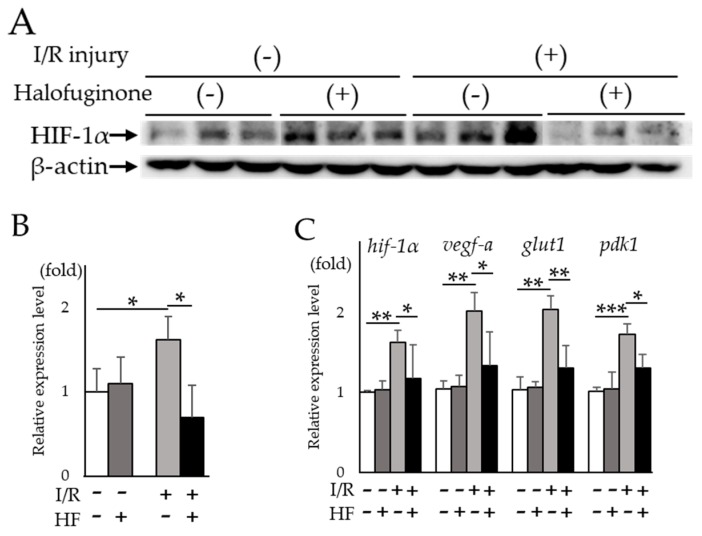
Suppression of increased HIF-1α and upregulated target genes by halofuginone administration in I/R retinas. (**A**) Western blotting for HIF-1α and β-actin in control or I/R retinas with or without halofuginone administration (*n* = 3). (**B**) Quantification of the blots showed that halofuginone administration suppressed increased HIF-1α protein expression. (**C**) *Hif-1α* and its representative target genes detected by qPCR (*n* = 3). Note that upregulated genes were suppressed by halofuginone administration. *Gapdh* was used as the internal control. Error bars indicate the standard deviation. HF; halofuginone. * *p* < 0.05, ** *p* < 0.01, *** *p* < 0.001, Student’s *t*-test.

**Figure 4 ijms-20-03171-f004:**
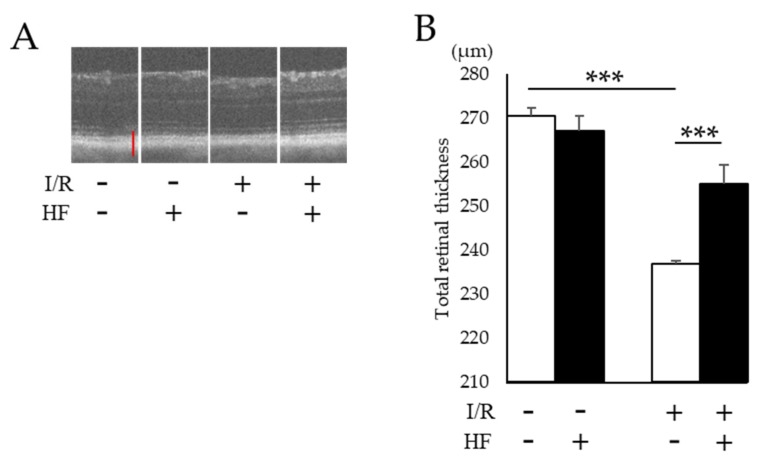
Evaluation of retinal morphology. (**A**) Representative optical coherence tomography (OCT) images of each group. Scale bar; 100 μm. (**B**) The average of total retinal thickness measured in OCT (*n* = 4). Note that the decreased total retinal thickness was prevented by halofuginone treatment post-I/R injury. (**C**) Representative H&E stained retinal sections. Scale bar; 100 μm. (**D**) The average of the total retinal thickness measured in H&E stained sections (*n* = 5). (**E**) The average of the inner retinal thickness in H&E stained retinas (*n* = 5). Note that the decrease of the thickness was found remarkably in inner retinal layers, whereas those changes were suppressed by halofuginone administration. HF; halofuginone. Error bars indicate the standard deviation. * *p* < 0.05, ** *p* < 0.01, *** *p* < 0.001, Student’s *t*-test.

**Figure 5 ijms-20-03171-f005:**
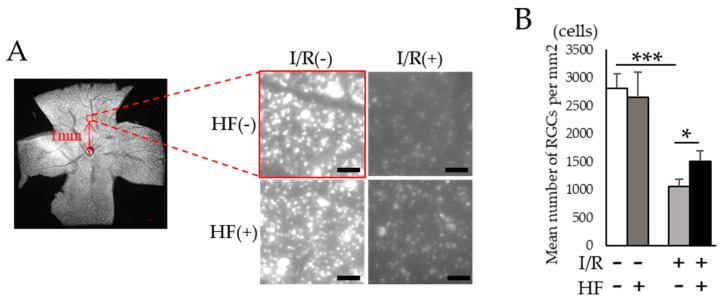
Fluorogold retrograde labeling of retinal ganglion cells (RGCs). (**A**) A representative quadrant retinal image with fluorogold-labeled RGCs. Red 200 μm square at 1 mm from optic disc head indicates the area for RGC densitometry (left). Magnified images for control and post-I/R retina with or without halofuginone treatment (right). Scale bars; 200 μm in quadrant retina, 50 μm in magnified images. (**B**) The quantification of RGC density for each group (*n* = 3). Note that decrease of RGCs was suppressed by topotecan administration. HF; halofuginone. Error bars indicate the standard deviation. * *p* < 0.05, *** *p* < 0.001, Student’s *t*-test.

**Figure 6 ijms-20-03171-f006:**
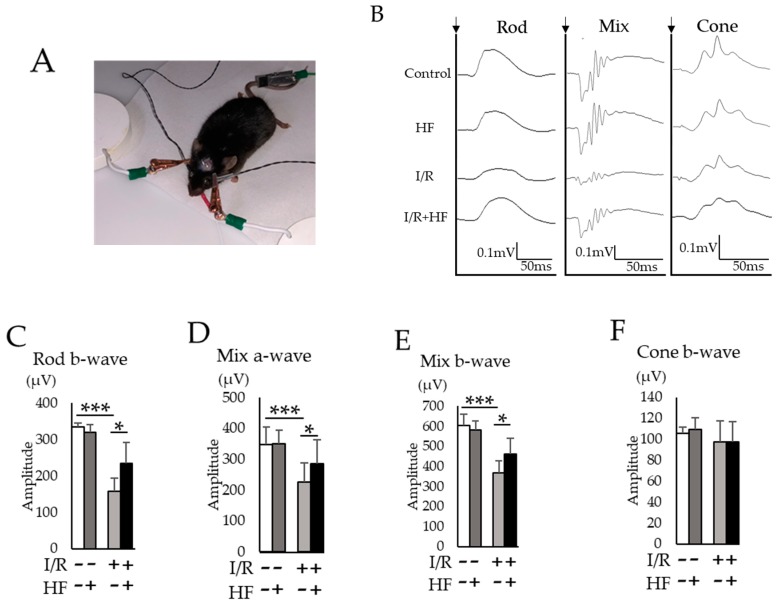
Retinal function evaluated with electroretinography (ERG). (**A**) A representative photograph of ERG recording. (**B**) Representative ERG waveforms for rod, mix, and cone conditions. Black arrows indicated the timing of the light stimulation. The averaged amplitudes were shown for rod b-wave (**C**), mixed a-wave (**D**), mixed b-wave (**E**), and cone b-wave (**F**) (*n* = 3–6). Note that decreased amplitudes in rod and mix conditions were suppressed by harofuginone administration. Error bars indicate the standard deviation. HF; halofuginone. * *p* < 0.05, *** *p* < 0.001, Student’s *t*-test.

**Figure 7 ijms-20-03171-f007:**
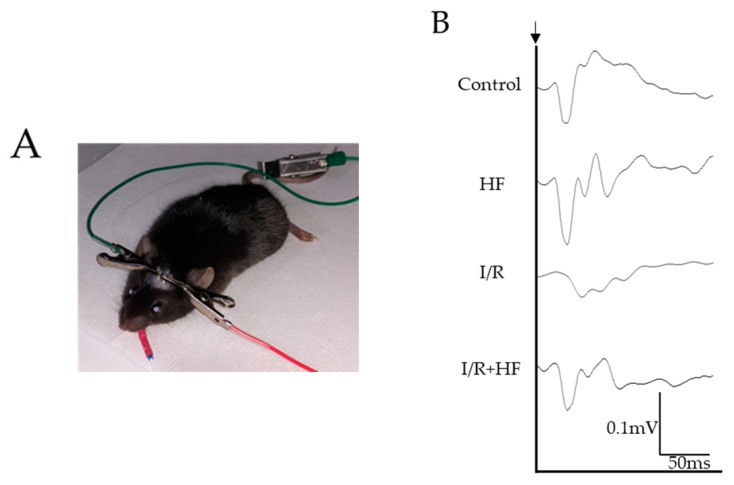
Evaluation of visual function detected with visual evoked potential (VEP). (**A**) A representative photograph of VEP recording. (**B**) Representative VEP waveforms from control and post-I/R retina with or without halofuginone treatment. A black arrow indicates the timing of the light stimulation. (**C**) The average of VEP amplitudes (*n* = 4). Note that decrease of VEP amplitude was suppressed by halofuginone administration. (**D**) The average of VEP implicit time (*n* = 4). The prolonged latencies by I/R injury were prevented in halofuginone-treated group. Error bars indicate the standard deviation. HF; halofuginone. * *p* < 0.05, ** *p* < 0.01, *** *p* < 0.001, Student’s *t*-test.

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
