# Peer review of "A Novel HIF Inhibitor Halofuginone Prevents Neurodegeneration in a Murine Model of Retinal Ischemia-Reperfusion"

_ijms, 2019, doi:10.3390/ijms20133171_

Round 1
Reviewer 1 Report
In the article entitled “A novel HIF inhibitor halofuginone prevents neurodegeneration in a murine model of retinal ischemia/reperfusion” by Kunimim et al., the authors screened natural products for their role in neuroprotection against I/R.
Major Concerns:
1) If the focus is glaucoma, why use a photoreceptor and RPE cell lines?
2) Is the dose of halofuginone in figure 2 physiological?
3) Figure 3—why does halofuginone increase HIF1a in non I/R samples?
4) Figure 4—it would be nice to see H&E stained retina also to support OCT data
5) Increasing IOP for only 30 minutes is uncommon to produce neurodegeneration—typically is for at least 60 min
6) Does halofuginone have a protein target to verify that it is inhibited its direct target or do the authors think HIF1a is the natural target?
7) Do the authors have a structure of this derivative?
Minor Concerns:
1) line 60, positive is misspelled
Author Response
We appreciate your careful review of our manuscript (ijms-527505). As requested, we resubmitted a revised copy of the manuscript. For emphasis, new text added to the manuscript is colored red, and the reviewer’s original questions are italicized and colored blue in this document. We believe that the revision addresses all comments raised by the reviewer.
Major Concerns:
1) If the focus is glaucoma, why use a photoreceptor and RPE cell lines?
The current study focuses on retinal ganglion cell (RGC) degeneration including glaucoma as pointed out. It is ideal to utilize RGCs for in vitro experiments; however, currently there exists no available RGC cell line. Thus, we have used 661W and ARPE19 cell lines derived from the retina. We added the description not to use RGCs in results and updated sentences as below:
2. Results
2.2. HIF inhibitory effects of halofuginone in vitro
We found that hydrangea extracts had the strongest effect to inhibit HIF activation in the natural product library (Table A2). In this plant species, febrifugine is known as a component to have pharmacological activities while this alkaloid has some adverse reactions including nausea. Then, we examined the HIF inhibitory effect of halofuginone, which is a synthetic derivative of febrifugine with fewer side effects, in 3 different types of cell lines (NIH/3T3, 661W a murine cone photoreceptor cell line, and ARPE19 a human retinal pigment epithelium (RPE) cell line). Although it is ideal to use retinal ganglion cells (RGCs) for validating the effect of halofuginone in neurodegeneration of inner retina, currently there exists no available RGC cell line. Thus, we have used 661W and ARPE19 cell lines derived from the retina in this study. In all these cell lines, the increase of HIF activity induced by CoCl2 was significantly suppressed by administration of halofuginone (Figure 1).
2) Is the dose of halofuginone in figure 2 physiological?
We assume that the dose used in the current in vitro study is physiological. According to a safety report of halofuginone (CVMP/643/99-Rev.1, EMEA 2003), maximal blood concentration of halofuginone (Cmax) reached to 4.12mg/ml (approximately 10nM) in new-born calves after bolus administration (0.1mg/kg). Based on the bioavailability in mice, more than 30nM of blood concentration is estimated with the one-day dose of the current in vivo experiment (0.2mg/kg). Although future actual evaluation of blood concentration is needed, the blood concentration is expected to reach to the dose shown in Figure 2 by continuous administration.
3) Figure 3—why does halofuginone increase HIF1a in non I/R samples?
There was no statistical difference between control and halofuginone-treated group in non I/R samples as shown in Figure 3B.
4) Figure 4—it would be nice to see H&E stained retina also to support OCT data
According to your suggestion, we performed an H&E staining and further confirmed histological neuroprotective effects of halofuginone against I/R damage (Figure 4C, D). In addition to the total retinal thickness, we evaluated changes of the inner retinal thickness in the H&E stained sections (Figure 4E). We updated sentences and figures as below:
2. Results
2.4. Improvement of RGC survival with halofuginone administration in post-I/R retinas
To assess the histological change of retinas after I/R injury, total retinal thickness was evaluated by optical coherence tomography (OCT). Total retinal thickness got thinner after I/R while those changes were prevented in halofuginone-treated mouse retinas (Figure 4A, B). We also evaluated the thickness with hematoxylin and eosin (H&E) stained retinal sections (Figure 4C). As consistent with the result in OCT, changes of the total retinal thickness were prevented by halofuginone administration (Figure 4D). The thickness of inner retinal layers between the inner retinal surface and the inner plexiform layer was significantly decreased with I/R damage. In contrast, the changes were suppressed in halofuginone-treated mouse retinas (Figure 4E).
Figure 4. Evaluation of retinal morphology. (A) Representative OCT images of each group. Scale bar; 100μm. (B) The average of total retinal thickness measured in OCT (n=4). Note that the decreased total retinal thickness was prevented by halofuginone treatment post-I/R injury. (C) Representative H&E stained retinal sections. Scale bar; 100μm. (D) The average of the total retinal thickness measured in H&E stained sections (n=5). (E) The average of the inner retinal thickness in H&E stained retinas (n=5). Note that the decrease of the thickness was found remarkably in inner retinal layers, whereas those changes were suppressed by halofuginone administration. HF; halofuginone. Error bars indicate the standard deviation. *p<0.05, **p<0.01,< span="">***p<0.001, Student’s t-test.
4. Materials and Methods
4.6. Evaluation of retinal thickness
Mouse total retinal thickness was evaluated using an OCT system (Envisu R4310, Leica, Germany). Mice were anesthetized with MMB and fixed in the system. Ten points of whole retinal thickness (250μm radius and every 36º centered upon the optic disc) were measured. The average number of them was evaluated as the value of the eye.
For H&E staining, eyes were enucleated from anesthetized mice, and frozen in the O.C.T. compound (Sakura Finetek Japan, Japan). Samples were sectioned by a cryostat (CryoStar, Thermo Fisher Scientific, United States). Slides were fixed in methanol for 5 minutes and dipped in hematoxylin solution for 90 seconds followed by a wash in water for 15 minutes. Then, slides were dipped in eosin solution for 30 seconds, and dipped 5 times in 100% ethanol and 3 times in Clear Plus (Falma, Japan) subsequently. Once slides were dried, mounting medium was applied and the slides were cover-slipped. Two points of retinal thickness (1mm from the optic disc) were measured in a section, and the averages of them were evaluated as the value of an eye.
5) Increasing IOP for only 30 minutes is uncommon to produce neurodegeneration—typically is for at least 60 min
Indeed, typical duration is 60-90 minutes for retinal I/R experiments as you pointed, but we confirmed a substantial neurodegeneration in the retina histologically and functionally with 30-minute IOP elevation in the current study. Another recent literature showed a significant murine retinal I/R damage with increased IOP for 45 minutes (Liu et al., Invest Ophthalmol Vis Sci. 2019).
6) Does halofuginone have a protein target to verify that it is inhibited its direct target or do the authors think HIF1a is the natural target?
As shown in Fig. 3B and C, halofuginone administration did not decrease the basal level of hif-1a mRNA. Additionally, we examined the change of HIF-1a protein expression by halofuginone with subsequent administration of a proteasome inhibitor MG-132 and found that halofuginone could decrease HIF-1a expression even under proteasome inhibition (data not shown). Taken together, halofuginone may have a direct target to inhibit HIF-1a expression independently from proteasome. Further studies are needed to conclude the mechanism of action of halofuginone.
7) Do the authors have a structure of this derivative?
The structures of halofuginone and other analogs derived from febrifugine have been reported (Jiang et al., Antimicrob Agents Chemother. 2005).
Minor Concerns:
1) line 60, positive is misspelled
Thank you for pointing out. We corrected the misspelling as below:
2. Results
2.1. Screening for novel HIF inhibitors
Our original library of natural compounds included 238 plants and food extracts. As the first screening, suppression of HIF activity was examined in the NIH/3T3 murine fibroblast cell line utilizing HIF luciferase dual assay. Cobalt chloride (CoCl2) was used to induce ectopic HIF activity and the suppressive effects of these products were evaluated (Table A1). Through the first screening, 78 candidates showed a stronger HIF suppressive effect compared to a positive control chemical topotecan. These compounds were further examined and statistically evaluated as the second screening (Table A2).
Reviewer 2 Report
The article by Kunimi et al. is an interesting one focusing on the importance of nutrition in the neuro-and eye- degenerative disorders. I am actually missing enough information about the description and importance of halofuginone in the introduction and discussion part. Thus, the introduction part might be improved.
I am also critical about the Fig 2. Why certain concentrations (100, 200, 300 nM) and duration (6 hours) of HF treatment were mentioned? Are there other concentrations being used and did not work? Why the figure is only with the 661w cells (mouse retinal cells), not ARPE19 (human retinal epithelial cells)?
A figure with the description of the treatment of mice along with electrophysiological examination will improve the quality of the article.
A description of the rod b-wave, mixed a-wave, mixed b-wave and cone b-wave will also improve the quality of the article. The description of the electrophysiological recordings in the method section should be clearly described.
Author Response
We appreciate your careful review of our manuscript (ijms-527505). As requested, we resubmitted a revised copy of the manuscript. For emphasis, new text added to the manuscript is colored red, and the reviewer’s original questions are italicized and colored blue in this document. We believe that the revision addresses all comments raised by the reviewer.
The article by Kunimi et al. is an interesting one focusing on the importance of nutrition in the neuro-and eye- degenerative disorders. I am actually missing enough information about the description and importance of halofuginone in the introduction and discussion part. Thus, the introduction part might be improved.
Thank you very much for your suggestion. We added a description about halofuginone in introduction. We updated sentences as below:
1. Introduction
In the current study, we examined 238 natural products to find out HIF inhibitory agents. Through the screening analysis, halofuginone (7-bromo-6-chlorofebrifugine hydrochloride) was found to have an effect of inhibiting HIF activity in vitro and in vivo. It was reported that halofuginone, an analogue of febrifugine, had therapeutic effects in fibrosis and other pathological conditions [14]. However, effects of halofuginone in eyes have not been reported yet. In this study, we examined the neuroprotective effect of halofuginone in a murine model of retinal ischemia-reperfusion (I/R).
I am also critical about the Fig 2. Why certain concentrations (100, 200, 300 nM) and duration (6 hours) of HF treatment were mentioned? Are there other concentrations being used and did not work? Why the figure is only with the 661w cells (mouse retinal cells), not ARPE19 (human retinal epithelial cells)?
Thank you for your important suggestion. According to your suggestion, we performed Western blotting for HIF-1a with lower HF concentrations (10 and 50nM) in 661W and ARPE19 cells under both CoCl2 and 1% oxygen conditions. These results indicate that HF has HIF-1a inhibiting effects with 100nM or higher concentrations. We updated sentences and figures as below:
2. Results
2.2. HIF inhibitory effects of halofuginone in vitro
We further examined protein expression of HIF-1α induced by two different conditions of CoCl2 and actual hypoxia 1% oxygen. Western blotting showed that both CoCl2- and 1% hypoxia- induced HIF-1α expression in 661W and ARPE19 cells was suppressed by administration of halofuginone in a dose-dependent manner (Figure 2). These results indicated that halofuginone inhibited ectopic protein expression of HIF-1α in pseudo and real hypoxic conditions.
Figure 2. Inhibition of HIF-1α protein expression by halofuginone administration in 661W and ARPE19 cells. Western blotting for HIF-1a under CoCl2 (A) or 1% oxygen conditions (B) in 661W cells, and under CoCl2 (C) or 1% oxygen conditions (D) in ARPE19 cells. CoCl2 was administrated at the concentration of 200μM, halofuginone was added at 10, 50, 100, 200, or 300nM simultaneously, and cells were incubated for 6 hours. The hypoxic condition was maintained in 1% O2 for 6 hours. Halofuginone inhibited HIF-1α protein expression in a dose-dependent manner in both conditions. HF; halofuginone.
A figure with the description of the treatment of mice along with electrophysiological examination will improve the quality of the article.
Thanks for your suggestion. We added photos of the ERG experiment as Figure 6A and of the VEP experiment as Figure 7A. We updated figures and legends as below:
Figure 6. Retinal function evaluated with ERG. (A) A representative photograph of ERG recording. (B) Representative ERG waveforms for rod, mix, and cone conditions. Black arrows indicated the timing of the light stimulation. The averaged amplitudes were shown for rod b-wave (C), mixed a-wave (D), mixed b-wave (E), and cone b-wave (F) (n=3-6). Note that decreased amplitudes in rod and mix conditions were suppressed by harofuginone administration. Error bars indicate the standard deviation. HF; halofuginone. *p<0.05, ***p<0.001, Student’s t-test.
Figure 7. Evaluation of visual function detected with VEP. (A) A representative photograph of VEP recording. (B) Representative VEP waveforms from control and post-I/R retina with or without halofuginone treatment. A black arrow indicates the timing of the light stimulation. (C) The average of VEP amplitudes (n=4). Note that decrease of VEP amplitude was suppressed by halofuginone administration. (D) The average of VEP implicit time (n=4). The prolonged latencies by I/R injury were prevented in halofuginone-treated group. Error bars indicate the standard deviation. HF; halofuginone. *p<0.05, **p<0.01, ***p<0.001, Student’s t-test.
A description of the rod b-wave, mixed a-wave, mixed b-wave and cone b-wave will also improve the quality of the article. The description of the electrophysiological recordings in the method section should be clearly described.
Thank you for pointing this out. We added a description for each wave in the results and details in the methods as below:
2. Results
2.5. Halofuginone administration protects I/R-damaged retina electrophysiologically
We used electroretinography (ERG) to assess the neuroprotective effect of halofuginone against retinal I/R damage. The amplitudes of a- and b-wave in three conditions of ERG (rod, mix, and cone) were evaluated (Figure 6A, B). In rod a-wave (Figure 6C) and mix a- (Figure 6D) and b-wave (Figure 6E), decreased amplitudes were significantly (p<0.05) improved by halofuginone administration. The amplitude in cone b-wave was not decreased by I/R (Figure 6F).Furthermore, we examined visual evoked potential (VEP) recordings (Figure 7A, B). The decreased amplitudes (Figure 7C) and extended latencies (Figure 7D) after I/R injury were also significantly (p=0.047, 0.006) improved in the halofuginone-treated group. These results suggested that halofuginone had a neuroprotective effect functionally against I/R damage.
4. Materials and Methods
4.8. Electrophysiological examination
In order to perform VEP recordings, two stainless steel pan-head screws (M1.0 x 6.0mm) used for electrodes were inserted 1.0mm in depth into the skull with hair shaved on the scalp around under MMB anesthesia before the I/R injury. These screws were placed over the bilateral primary visual cortex (1.5mm anterior and 1.5mm lateral to lambda). Mice were dark-adapted overnight before the electrophysiological examination. Full field ERG and VEP were recorded at 7days after I/R using an acquisition system (PuREC, Mayo, Japan) with an LED stimulator in a Ganzfeld dome. Mice were anesthetized and dropped mydriatic agent to both eyes. For ERG recordings, custom active electrodes were placed with contact lens (Mayo, Japan) and a reference electrode was placed in mouth. Three ERG stimulation intensities were used throughout recordings; 0.02cds/m² 4 times for rod, 50cds/m² one time for mix, and 20cds/m² 32 times for cone conditions. For VEP recordings, the active electrodes were inserted into the screws of skull and the reference electrode was placed in the mouth. A clipping electrode to the tail served as a ground. All recordings were measured in photopic adaptation status with 1Hz white flashlights (3cds/m²). Responses were arithmetically averaged with up to 64 records. The signals were 25Hz low pass filtered. Each animal was recorded twice for both eyes. The amplitudes were taken with the first negative peak (P1-N1).
Round 2
Reviewer 1 Report
This paper has been much improved